# Structure of the *Arabidopsis* guard cell anion channel SLAC1 suggests activation mechanism by phosphorylation

Yawen Li[1,5], Yinan Ding [1,2,5], Lili Qu[1,2], Xinru Li[1], Qinxuan Lai[1], Pingxia Zhao[3], Yongxiang Gao[4], Chengbin Xiang[3], Chunlei Cang [1,2✉], Xin Liu [1✉] & Linfeng Sun [1✉]

Stomata play a critical role in the regulation of gas exchange and photosynthesis in plants. Stomatal closure participates in multiple stress responses, and is regulated by a complex network including abscisic acid (ABA) signaling and ion-flux-induced turgor changes. The slow-type anion channel SLAC1 has been identified to be a central controller of stomatal closure and phosphoactivated by several kinases. Here, we report the structure of SLAC1 in *Arabidopsis thaliana* (*At*SLAC1) in an inactivated, closed state. The cytosolic amino (N)-terminus and carboxyl (C)-terminus of *At*SLAC1 are partially resolved and form a plug-like structure which packs against the transmembrane domain (TMD). Breaking the interactions between the cytosolic plug and transmembrane domain triggers channel activation. An inhibition-release model is proposed for SLAC1 activation by phosphorylation that the cytosolic plug dissociates from the transmembrane domain upon phosphorylation, and induces conformational changes to open the pore. These findings facilitate our understanding of the regulation of SLAC1 activity and stomatal aperture in plants.

[1] Department of Neurology, the First Affiliated Hospital of USTC, MOE Key Laboratory for Membraneless Organelles and Cellular Dynamics, Hefei National Laboratory for Physical Sciences at the Microscale, CAS Centre for Excellence in Molecular Cell Science, Biomedical Sciences and Health Laboratory of Anhui Province, Division of Life Sciences and Medicine, University of Science and Technology of China, Hefei 230027, China. [2] Institute on Aging and Brain Disorders, the First Affiliated Hospital of USTC, CAS Key Laboratory of Innate Immunity and Chronic Disease, University of Science and Technology of China, Hefei 230027, China. [3] School of Life Sciences, Hefei National Laboratory for Physical Sciences at the Microscale, University of Science and Technology of China, The Innovation Academy of Seed Design, Chinese Academy of Sciences, Hefei 230027, China. [4] High-End Cryo-EM Platform, Core Facility Center for Life Sciences, University of Science and Technology of China, Hefei 230027, China. [5] These authors contributed equally: Yawen Li, Yinan Ding. ✉email: ccang@ustc.edu.cn; lx023@ustc.edu.cn; sunlf17@ustc.edu.cn

Gas exchange between plants and the surrounding environment, such as the uptake of $CO_2$ and release of $O_2$ in photosynthesis or the loss of water vapor in transpiration, occurs mainly through the microscopic pores in the epidermis of leaves or young stems, named stomata[1]. The stomata is surrounded by a pair of specialized, kidney-shaped guard cells, which integrate various abiotic or biotic signals to fine tune the stomatal aperture through its turgor change. The increase in guard cell turgor triggers stomatal opening, while the decrease prompts stomatal closure. Stomatal closure represents a critical self-protection response for the plants to a variety of stress stimuli, for instance, high $CO_2$ or ozone concentration, drought, cold temperature, or pathogen invasion[2]. During stomatal closure, anion channels are activated and the efflux of anions from guard cells, especially chloride ($Cl^-$) and nitrate ($NO_3^-$), causes membrane depolarization, which in turn activates the outward-rectifying potassium ($K^+$) channel and facilitates $K^+$ efflux. Consequently, the cell turgor decreases and the ensuing loss of water leads to the guard cell shrinkage and stomatal aperture reduction[3].

Slow anion channel 1 (SLAC1), an S-type (slow) anion channel of the SLAC/SLAH family which is preferentially expressed in guard cells, has been identified to be mainly responsible for the stress-induced anion efflux and thus playing a pivotal role mediating stomatal closure[4,5]. It is targeted by a multitude of stimulating factors, such as $CO_2$, $O_3$, light, and plant hormones like abscisic acid (ABA) which is a principal regulator of stress responses[6–10]. The regulation of SLAC1 by high $CO_2$ and ABA has been the most studied and multiple activation or inactivation pathways have been revealed[11]. For high $CO_2$, after diffusion into the guard cell, it is converted to $HCO_3^-$ firstly. $HCO_3^-$ can lead to the activation of multiple protein kinases that function in parallel, like the SnRK2 (Snf1-related protein kinase 2) family member OST1 or the LRR-RLK (leucine-rich repeat receptor-like protein kinase) family member GHR1, which phosphorylates SLAC1 and somehow opens the channel[8,12–14]. Meanwhile, it also binds directly to the transmembrane domain of SLAC1 and enhance the channel activity[15]. In the ABA signaling pathway, SLAC1 can be activated via either the $Ca^{2+}$-independent or $Ca^{2+}$-dependent pathway. In the $Ca^{2+}$-independent pathway, ABA binds to its receptors of the RCAR/PYR1/PYL family, then interacts with the PP2C protein phosphatase family members like ABI1 and releases its inactivation of OST1, which in turn activates SLAC1 through phosphorylation[8,9]. In the $Ca^{2+}$-dependent pathway, the elevated cytosolic calcium associated with ABA signaling either directly activates multiple calcium-dependent protein kinases like CPK3, CPK6, CPK21, and CPK23, or binds to the calcineurin B-like (CBL) calcium sensors like CBL1 or CBL9 and then activates the CBL interacting protein kinases (CIPKs) such as CIPK23[16–20]. Similar to OST1 or GHR1, the activated CPKs or CIPKs can phosphorylate SLAC1 and lead to stomatal closure. The protein phosphatases like ABI1 or ABI2 function as negative regulators of SLAC1 by dephosphorylating the protein kinases and inhibit their activities, and by directly dephosphorylating SLAC1[16–18].

Using in vitro assays or in vivo examinations with *Xenopus* oocytes or plant guard cells, the mechanism for how SLAC1 activity is controlled by phosphorylation/dephosphorylation has been deeply investigated and a series of serine/threonine residues have been revealed to undergo modifications that are essential to the channel activation. All of these sites are located in the cytosolic N-terminal or C-terminal tail, like S59, S86, and S120 in the N-terminus and S543 in the C-terminus of *Arabidopsis* SLAC1 (*At*SLAC1)[8,9,16,17,20,21]. Although the in vitro data shows that a specific type of kinase can phosphorylate most of these sites, the in vivo results suggest that there are strong preferences in the

phosphorylation site for different kinases[17]. S59 is critical for *At*SLAC1 activation by the calcium-dependent kinase CPKs, while S120 is required for activation by the calcium-independent kinase OST1. Structural studies of a bacterial homolog of SLAC1 from *Haemophilus influenzae* (*Hi*TehA) and a grass SLAC1 from *Brachypodium distachyon* (*Bd*SLAC1) have brought insightful understandings of the channel architecture and pore gating[21,22]. However, due to lacking the substantial N- or C-terminus of *At*SLAC1 in *Hi*TehA, or the intrinsic flexibilities of both termini in *Bd*SLAC1, which lead to an invisible cytosolic domain in the structure, the molecular mechanism for SLAC1 regulation by the kinase/phosphatase pairs remains elusive.

In this study, we report the structure of a SLAC1 mutant from *Arabidopsis thaliana* (*At*SLAC1 S59A) in an inactivated state at 2.7 Å resolution as determined by cryo-electron microscopy (cryo-EM) single-particle analysis. A large part of the cytosolic domain composed of the N- and C-terminus is clearly identified and forms a plug-like structure which interacts with the transmembrane domain (TMD) and blocks the anion pore, hence inhibiting channel opening. We characterized the channel activity in the HEK293 cells, where the protein was expressed and purified for structure determination, and found that by breaking the interactions between the cytosolic plug and TMD, *At*SLAC1 could be directly activated without the key site phosphorylation. Based on these structural and electrophysiological analysis results, we propose an inhibition-release-triggered pore dilation mechanism for the SLAC1 activation upon kinase phosphorylation.

## Results

**Protein expression and electrophysiological characterization of *At*SLAC1 in the HEK293 cell system**. HEK293 is a cell line derived from human embryonic kidney and has been extensively used as a vehicle for recombinant protein expressions, especially for eukaryotic membrane proteins. To test protein expression in the HEK293 cell system, we subcloned the wild type (WT) cDNA of *At*SLAC1 into a pCAG vector with a FLAG tag sequence for the affinity purification, and transfected to the HEK293F suspension cell cultures for transient expression. The protein had a considerable yield and behaved well in the size exclusion chromatography (Supplementary Fig. 1a). We then proceeded to prepare cryo-EM samples and tried to determine its structure using single-particle analysis. Although the particles were well-dispersed in the sample under EM imaging and clear features were identified with picked particles after 2D classification, the resolution of 3D reconstruction could not be improved beyond 6 Å, precluding secondary structure assignment and model building (Supplementary Fig. 1b–d). The problem still persisted with larger data sets processed or different approaches to prepare the protein sample.

Meanwhile, we attempted to characterize the channel activity in the HEK293T cell system. The same vector for *At*SLAC1 expression was transfected to the adherent cell culture, together with a vector encoding the green fluorescent protein (GFP) as an indicator of successful transfection. Whole-cell voltage clamp recordings were performed on cells with green fluorescence 48 h after transfection. As a negative control, no apparent currents were detected in cells expressing the empty pCAG vector (Supplementary Fig. 1e). Surprisingly, robust anion currents were recorded when voltage was applied to cells expressing the WT *At*SLAC1, similar to the S-type anion currents observed in the electrophysiological measurements using the guard cell protoplasts or *Xenopus* oocytes (Fig. 1a). However, this is distinct from the recordings using *Xenopus* oocyte, in which apparent currents are only detected by coexpression of *At*SLAC1 with the plant protein kinase or expression of the chimeric SLAC1 and OST1

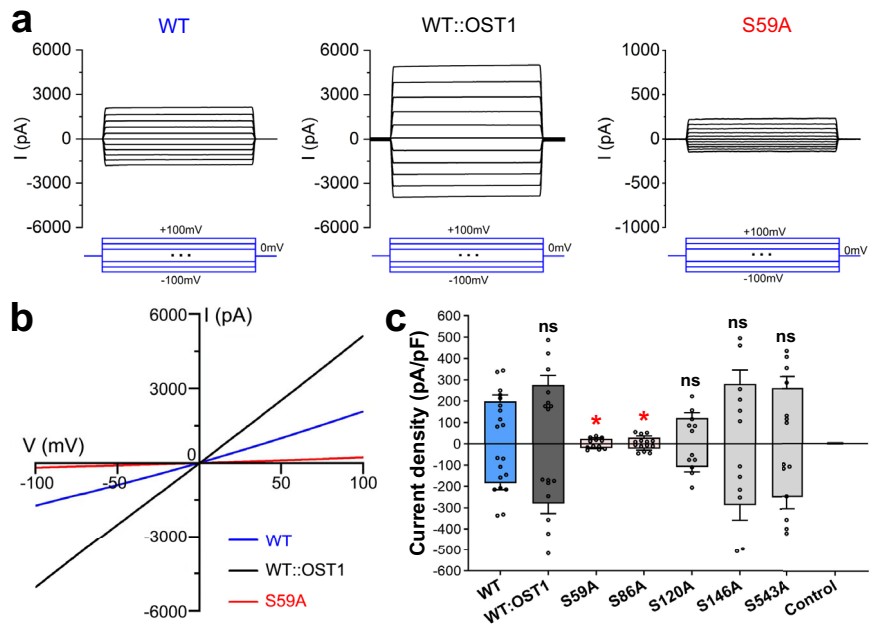

**Fig. 1 Electrophysiological characterization of *At*SLAC1. a** Representative traces of the whole-cell voltage clamp recording (500 ms step pulses, −100 mV to +100 mV, 20 mV step; $V_{holding} = 0$ mV) from HEK293T cells expressing the wild type *At*SLAC1 (WT), the *At*SLAC1 and OST1 fusion construct (WT::OST1), or the S59A mutant. **b** Representative whole-cell currents recorded using a ramp protocol (−100 mV to +100 mV in 1 s; $V_{holding} = 0$ mV) in HEK293T cells expressing WT *At*SLAC1, WT::OST1 fusion construct or S59A mutant. Compared to the WT *At*SLAC1 and the fusion construct, the S59A mutant shows a much impaired current under the same voltage. **c** Average outward and inward current densities measured at +100 mV and −100 mV in the ramp protocol used in (**b**), respectively, for the WT *At*SLAC1, the fusion construct WT::OST1, the phosphorylation site mutants, and the empty pCAG vector as a control. Of the five phosphorylation site mutants, S59A and S86A exhibit significantly reduced current densities. Independent experiments were repeated for each construct (WT, *n* = 10; WT::OST1, *n* = 8; S59A, *n* = 6; S86A, *n* = 7; S120A, *n* = 6; S146A, *n* = 6; S543A, *n* = 7; Control, *n* = 6). Significances were determined using one-way ANOVA with Dunnett's multiple comparisons test. *$P < 0.05$, ns = not significant for WT versus mutants. Data are represented as mean ± S.E.M.

fusion construct[8,9,21,22]. We also generated a construct of *At*SLAC1 directly fused at its C-terminus with the calcium-independent kinase OST1 (named SLAC1::OST1). The channel activity of this fusion construct was increased, and the average inward or outward current density measured at −100 mV or +100 mV was larger than that of the WT *At*SLAC1 by 30% (Fig. 1a–c). Together, these results suggest that when expressed in the HEK293 cells, *At*SLAC1 is robustly activated, and its activity can be further enhanced by the introduction of its kinase OST1.

To find out whether *At*SLAC1 is activated through phosphorylation when expressed alone in the HEK293 cells, we submitted the purified WT protein to the mass spectrometry analysis. Remarkably, five sites were identified to be phosphorylated, including S59, S86, S120, and S146 of the N-terminus and S543 of the C-terminus (Supplementary Fig. 1f and 2a). Of these sites, S59, S86, and S120 have been identified to be phosphorylated by OST1 or CPKs, and are essential to *At*SLAC1 phosphoactivation[17,20,23]. The other two have also been identified to be phosphorylated when SLAC1 is coexpressed with OST1[21]. It shows that when expressed alone, *At*SLAC1 is phosphorylated by some endogenous protein kinase(s) present in the HEK293 cells, which mimics the phosphoactivation process of the anion channel. However, compared to the previously reported 14 sites identified to be phosphorylated in the SLA-C1::OST1 chimera[21], *At*SLAC1 is phosphorylated to a lesser extent and is partially activated, which explains the lower average current density observed for the WT channel than the fusion construct. Staurosporine is a potent, broad-spectrum inhibitor of protein kinases[24]. Upon pretreatment with staurosporine, the anion current densities for cells transfected with the WT *At*SLAC1 channel were reduced in a dose-dependent manner (Supplementary Fig. 1g), which further supports the idea that *At*SLAC1 is activated through

phosphorylation by endogenous kinases in the HEK293 cells. To test the roles of the five identified phosphorylation sites in *At*SLAC1 activation, we mutated the serine residues to alanine, respectively, and examined the anion currents when expressed alone in HEK293T cells. Of all the five sites, S59 and S86 appear to be the most critical ones, as once mutated to alanine, the inward or outward current or the average current density measured at −100 mV or +100 mV was reduced dramatically for either mutant (Fig. 1a–c). For the S120A mutant, the current densities were slightly reduced. For the S146A and S543A mutants, the current densities were comparable to, or slightly higher than the WT *At*SLAC1 (Fig. 1c). The impaired anion channel activity for the *At*SLAC1 S59A mutant in the HEK293 cell system is consistent with previous electrophysiological results using the *Xenopus* oocytes, and notably, S59 has been identified to be preferentially required for *At*SLAC1 activation in oocytes by the calcium-dependent kinase CPKs[17]. The S86 site has also been shown to play an important role in SLAC1 activation, as the SLAC1::OST1 S86A mutant had much-reduced currents when tested in the oocytes[21].

**Overall architecture of the *At*SLAC1 S59A mutant.** Taken together the protein purification and electrophysiological analysis results, the HEK293 cell system provides us a good platform for structural and functional studies of the SLAC1 channel. As mentioned above, the resolution for the 3D reconstruction of the WT *At*SLAC1 could not be further improved, possibly due to the partially activation of *At*SLAC1 in the HEK293 cells as shown by electrophysiological analysis and the resulting heterogeneity in protomer conformations. We turned to the phosphorylation site mutants which had impaired anion currents and might be

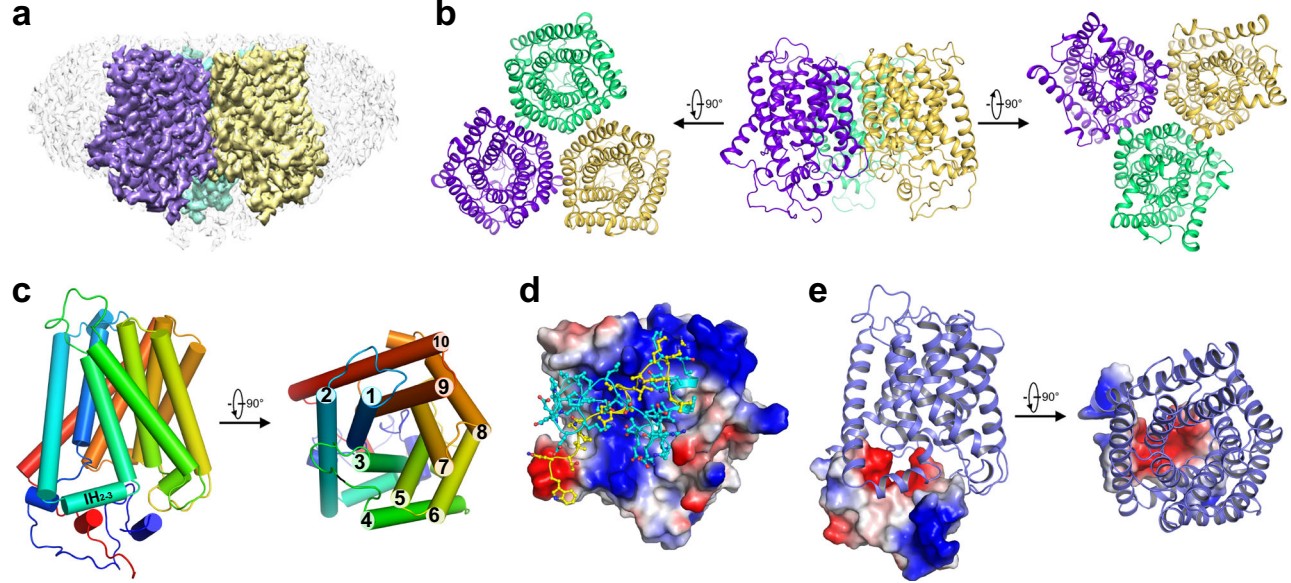

**Fig. 2 Structure determination of the *At*SLAC1 S59A mutant. a** A side view of the EM density for the *At*SLAC1 S59A mutant at 2.7 Å resolution. Densities corresponding to the three subunits are color-coded, and the surrounding micelle is shown in light gray. The density maps are generated by Chimera[39]. **b** Three different views are presented here for the structure model of *At*SLAC1 S59A mutant. **c** Overview of the subunit structure of *At*SLAC1 S59A mutant. The ten transmembrane segments (TMs) are organized in two layers, with the inner layer forming the anion conducting pore. TM2 and TM3 is linked by a short juxtamembrane helix, named IH$_{2-3}$. **d** The surface electrostatic potential of the transmembrane domain of *At*SLAC1 viewed from the cytoplasmic side. The N- and C-terminus of *At*SLAC1 are shown in sticks, colored cyan and yellow, respectively. **e** The surface electrostatic potential of the cytosolic domain of *At*SLAC1 shown in a side view (left) and top view from the extracellular side (right). The electrostatic potential is highly negative near the pore for the cytosolic domain, and complements to the positive potential of the transmembrane domain in this region.

stabilized in an inactivated conformation, specifically, the S59A mutant, which has been shown extensively to trim down *At*SLAC1 activation and stomatal closure and also has notable reduced currents in the HEK293 cells. We purified the *At*SLAC1 S59A mutant using the same procedure as the wild-type protein. The protein yield and behavior in gel filtration were similar to the WT *At*SLAC1 (Supplementary Fig. 3a). The purified protein was applied to cryo-EM sample preparation and imaging, which showed well-dispersed particles suitable for data collection (Supplementary Fig. 3b). Finally, we were able to obtain an EM density map with an overall resolution of 2.7 Å according to the gold-standard Fourier shell correlation (FSC) criterion, which allowed us to build an atomic model for the *At*SLAC1 S59A mutant (Fig. 2a, Supplementary Fig. 3c–g, Supplementary Fig. 4). Detailed data collection and processing procedures can be found in the methods section.

Similar to *Hi*TehA and *Bd*SLAC1, *At*SLAC1 forms a homotrimer that is tightly packed in the transmembrane domain (Fig. 2b)[21,22]. Each monomer contains ten transmembrane segments (TMs) that are arranged in two pentagonal layers, with TM1/3/5/7/9 in the inner layer and TM2/4/6/8/10 in the outer layer (Fig. 2c). The inner layer contributes to the anion pore formation. Notably, the cytosolic linker between TM2 and TM3 forms a short helix lying almost parallel to the membrane (named IH$_{2-3}$ hereafter). Structural alignment reveals that *At*SLAC1 and *Bd*SLAC1 have an almost identical architecture in the transmembrane domain, with a root-mean-square deviation (RMSD) of 0.81 Å (273 Cα atoms aligned), except a slight shift in the C-terminus of TM10 (Supplementary Fig. 5a).

The full-length *At*SLAC1 protein contains 556 residues. In the built atomic model, the N-terminus starts at residue 150, and the C-terminus ends at residue 517. The first 149 and the last 39 residues were unmodelled due to vague EM densities. All of the modeled residues are faithfully assigned due to the high resolution. Compared with *Bd*SLAC1, the N-terminus of

*At*SLAC1 structure is extended by 32 amino acids and the C-terminus is elongated by 15 amino acids, offering a privilege to dissect more structural information of the cytosolic domain. Remarkably, both tails extend towards the pore and closely pack to the inner surface of the transmembrane domain (Fig. 2d). Together, they form a plug-like structure that inserts into the inner side of the pore (Fig. 2e). To be noticed, the intracellular surface of the transmembrane domain has a positive electrostatic potential (Fig. 2d), while the plug has a complementary negative electrostatic potential facing the TMD, suggesting strong electrostatic interactions between the TMD and the plug (Fig. 2e).

**AtSLAC1 is in an inactivated, closed state.** Though assembled into a trimeric structure, each subunit of *At*SLAC1 functions as an independent anion pore. The extracellular and intracellular surface of the pore both contain several positively charged residues, like K211, K290, K347, and K461 at the extracellular surface, and R263, H260, H387, and K440 at the intracellular surface (Supplementary Fig. 5b). Consequently, both surfaces have a positive electrostatic potential, favorable for the anion passaging.

Pore radius analysis of the *At*SLAC1 structure reveals that the upper half of the channel is wide open to the periplasmic side of the cell, whereas the lower half is completely sealed (Fig. 3a). Thus, the *At*SLAC1 S59A mutant is in an inactivated, closed state in our determined cryo-EM structure. It also reinforces the statement that conformational changes have to occur in the transmembrane domain to enlarge the pore upon channel activation, especially for the lower half of the pore, besides protein phosphorylation. A close examination of the pore-lining residues reveals that three ring-shaped layers of hydrophobic residues are located along the anion path. Especially, each layer contains a phenylalanine residue which has a large aromatic ring, F276, F450, and F266 (Fig. 3b). We examined the channel activities of these phenylalanine mutants in the HEK293 cells. When expressed alone, the inward current densities measured at

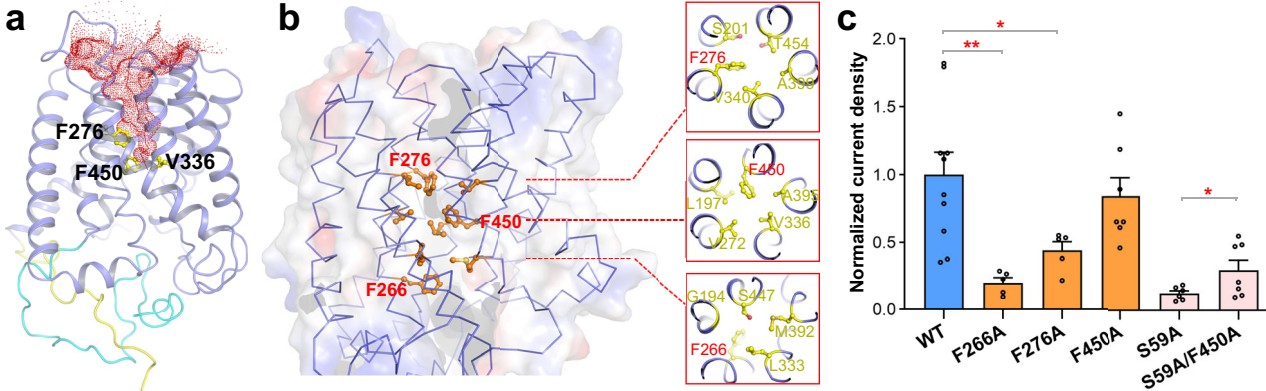

**Fig. 3 AtSLAC1 S59A structure is in a nonconductive, inactivated state. a** Pore depiction of the AtSLAC1 S59A structure. The anion conduction pore, calculated by HOLE2[40], is shown as red mesh. The structure is shown in a cartoon model, with the transmembrane domain colored slate blue, the N-terminus colored cyan, and the C-terminus colored yellow, respectively. The upper half of the channel is open to the periplasmic side of the cell, whereas the lower half is completely sealed. **b** Three ring-shaped layers of hydrophobic residues located along the anion path are shown in sticks. Each layer contains a phenylalanine residue, F276, F450 and F266. **c** Normalized inward current densities measured at −100 mV for the phenylalanine residue mutants compared to the WT AtSLAC1 characterized in the HEK293T cells. Independent experiments were repeated for each construct (WT, $n = 10$; F266A, $n = 5$; F276A, $n = 5$; F450A, $n = 7$; S59A, $n = 6$; S59A/F450A, $n = 7$). Significances were determined using one-way ANOVA with Dunnett's multiple comparisons test for the single phenylalanine mutants, and unpaired $t$-test between the S59A and S59A/F450A mutants. $*P = 0.0354$ for the F276A mutant; $**P = 0.0296$ for the S59A/F450A mutant; $**P = 0.0023$ for the F266A mutant. Data are represented as mean ± S.E.M.

−100 mV for the F266A and F276A mutants were reduced by 80% and 50%, respectively (Fig. 3c). For F450, which locates at the narrowest site of the pore facing the extracellular side (Fig. 3a), once mutated to alanine, the inward current density was barely affected (Fig. 3c). Combining F450A with the phosphorylation site mutation S59A, the S59A/F450A double mutant increased the inward current density by almost two folds compared to the S59A single mutant, while still remained much lower than the WT AtSLAC1 (Fig. 3c). These results suggest that the three phenylalanine residues may have distinct roles during channel activation, and it cannot remarkably dilate the pore by simply mutating the large aromatic residue to alanine. On the contrary, mutating F266 and F276 may impair the conformational changes needed to open the channel and lead to the much reduced inward currents. Notably, all three phenylalanine residues are conserved between BdSLAC1 and AtSLAC1 and adopt almost identical configurations in the structures (Supplementary Fig. 5c and Supplementary Fig. 6). Our electrophysiological results for the pore-lining phenylalanine mutants of AtSLAC1 are consistent with the results for the same mutants of BdSLAC1: replacements of the phenylalanine residue by alanine decrease the current in phosphorylated channels, while increase the current in unphosphorylated or less phosphorylated channels[21]. Unequivocally, the imposed constraints from these pore-lining phenylalanine residues have to be released upon channel activation, while the detailed scenario in structural rearrangements remain elusive.

Another notable feature is that in each of the pore-forming transmembrane segment, a "helix breaker" which tends to disrupt the helical structure, either a glycine or proline residue, is identified near the middle of the helix, namely G196 in TM1, P270 in TM3, G337 in TM5, P397 in TM7 and P451 in TM9 (Supplementary Fig. 5d). Kinked helices are commonly present in channels or transporters that undergo conformational changes to regulate substrate transport. The "helix breakers" may be anchors around which the pore-forming helices rotate or twist to regulate pore opening. The P451A or P451G mutants have been previously shown to have notably impaired currents in Xenopus oocytes[21]. We also tested the expression and electrophysiology of P451A mutant in the HEK293 cell system. Compared with the

WT AtSLAC1, the P451A mutant had a comparable protein yield and solution behavior in gel filtration (Supplementary Fig. 5e). However, the anion currents were almost completely abolished for the P451A mutant when expressed alone in the HEK293T cells (Supplementary Fig. 5f, g), similar to the results in the oocytes[21], suggesting essential roles for the kinking residue during channel activation.

**Breaking the interactions between the intracellular plug and transmembrane domain triggers AtSLAC1 activation.** As aforementioned, the cytosolic domain formed by the N-terminal and C-terminal tails closely packs to transmembrane domain of AtSLAC1. Specially, a loop region in the N-terminus inserts to the inner side of the pore and forms a plug-like structure (Fig. 4a). Scrutinizing the interface reveals that three pairs of residues mainly mediates the interactions between the transmembrane helices and the N- or C-terminus. E164 in the N-terminus forms a hydrogen bond with Y448 in TM9, and D166 in the N-terminus forms an ionic interaction with R263 in TM3. Besides, R263 also interacts with D507 in the C-terminus (Fig. 4a). To check the roles of these interacting pairs, we generated a series of alanine mutations of these residues with or without the S59A mutation, and analyzed their electrophysiological behaviors in the HEK293T cells. In the absence of S59A mutation, the E164A, D166A, and R263A mutants all showed dramatically enhanced inward currents densities measured at −100 mV compared to the WT AtSLAC1, indicating a more activated anion channel besides phosphorylation (Fig. 4c). The inward current densities for the Y448A and D507A mutants were also slightly higher than the wild-type channel. In combination with the S59A mutation, the S59A/E164A, S59A/D166A, S59A/R263A, and S59A/D507A double mutants all increased the anion currents to certain extents compared to the S59A single mutant (Fig. 4d). S59A/E164A, S59A/R263A, and S59A/D507A were the most profound ones which showed currents comparable to, or even larger than the WT AtSLAC1 (Fig. 4c, d). Although the Y448A single mutation retained the activated anion currents, the S59A/Y448A double mutation did not exhibit apparent anion currents. Taken together, breaking the interactions between the intracellular plug and

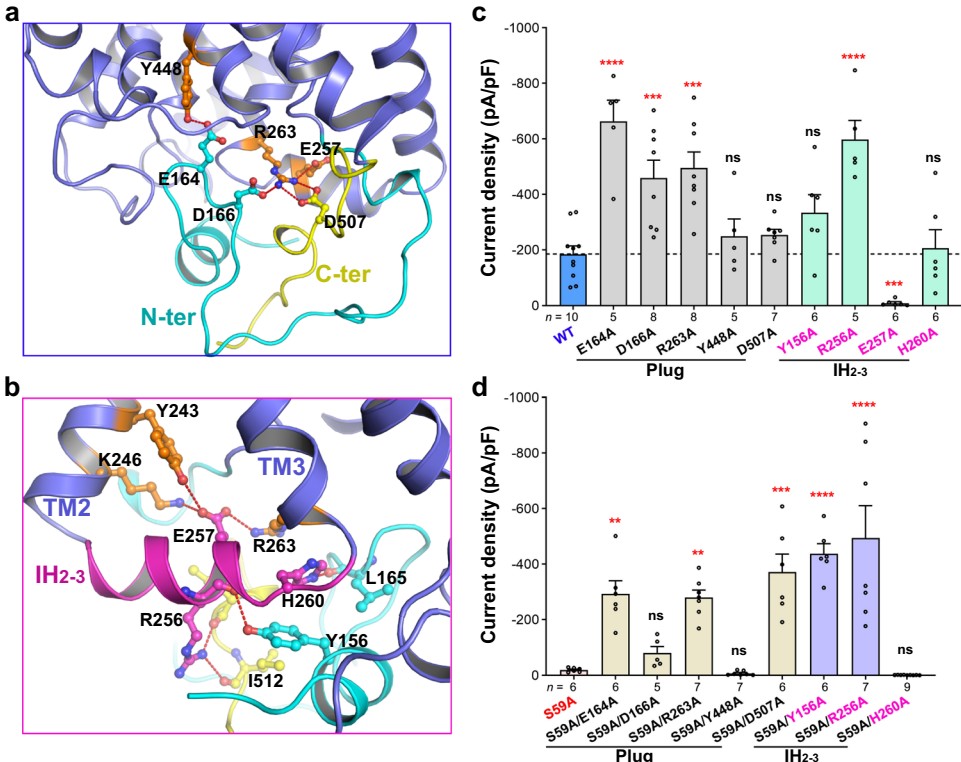

**Fig. 4 Interactions between the cytosolic and transmembrane domain of *At*SLAC1. a** Interactions between the pore-forming transmembrane helices and the cytosolic plug formed by the N- and C-terminal tails. **b** Residues from the $IH_{2-3}$ linker interacts with both the transmembrane segments and the cytosolic tails. $IH_{2-3}$ is colored magenta. **c** Inward current densities for the alanine mutants which are involved in the interactions between the plug and the transmembrane helices, or between $IH_{2-3}$ and its surroundings, measured at −100 mV, respectively, in the HEK293T cells. **d** Inward current densities for the alanine mutants in combination with S59A measured at −100 mV in the HEK293T cells. For all the constructs examined in **c** and **d**, independent experiments were repeated and the number of cells used ($n$) for each construct were indicated under the column. Significances were determined using one-way ANOVA with Dunnett's multiple comparisons test. **$P < 0.01$, ***$P < 0.001$, ****$P < 0.0001$, ns = not significant for WT versus mutants in **c**, or S59A versus the double mutants in **d**. Data are represented as mean ± S.E.M.

TMs can trigger *At*SLAC1 activation even in the absence of the key site phosphorylation, and the cytosolic plug observed in the *At*SLAC1 S59A structure, especially the N-terminal tail, may serve as a pore blocker prior to phosphoactivation.

Besides interacting with the transmembrane segments of *At*SLAC1, the N- or C-terminal tail also interacts with the aforementioned helical linker between TM2 and TM3, $IH_{2-3}$ (Fig. 4b). Specifically, hydrogen bonding interactions are found between the hydroxyl group of Y156 in the N-terminus and the main-chain carbonyl group of R256 in $IH_{2-3}$, the main-chain carbonyl group of L165 in the N-terminus and the imidazole ring of H260 in $IH_{2-3}$ (Fig. 4b). Meanwhile, the side chain of R256 interacts with the main-chain carbonyl groups of both I510 and I512 in the C-terminal tail (Fig. 4b). We generated alanine mutations of the interacting residues and examined their channel activities in HEK293 cells. For the single point mutant, while H260A had little effect to the current densities, Y156A and R256A showed much more increased activities than the WT *At*SLAC1 (Fig. 4c). For the double mutants, while S59A/H260A had hardly any detectable currents, both S59A/Y156A and S59A/R256A mutants showed remarkable inward current densities measured at −100 mV, much larger than that of the WT *At*SLAC1 when expressed alone in HEK293 cells (Fig. 4d). These results suggest that *At*SLAC1 can also be activated by breaking the interactions between $IH_{2-3}$ and the N- or C- terminal tails and further support the inhibition role of the cytosolic domain. Notably, a negatively charged residue in $IH_{2-3}$, E257, interacts with residues from both TM2 (Y243 and K246) and the pore-

forming helix TM3 (R263) (Fig. 4b). By mutating E257 to alanine, the anion channel activity was almost completely abolished (Fig. 4c). As mentioned above, upon channel activation, conformational changes have to occur, especially in the lower half of the transmembrane segments, to dilate the pore. $IH_{2-3}$ may play a critical role during this process as bridging the changes induced by the release of the cytosolic plug to the pore-forming helices and the outer layer TMs.

In the *At*SLAC1 structure, the C-terminal tail forms extensive interactions with the N-terminal tail, like hydrogen bonding or ionic interactions (Supplementary Fig. 7a). Notably, a speculated phosphorylation site in as previously reported[20], T513, interacts with the side chain of R155 through its main-chain carbonyl group (Supplementary Fig. 7a). While no apparent interactions are observed between the side chain of T513 and other residues, T513 is surrounded by several charged residues, like R155, E172, D173, K514, R515, and K516 (Supplementary Fig. 7b). The phosphomimetic mutant T513E did exhibit higher anion currents compared to the WT *At*SLAC1 when expressed alone in the HEK293 cells (Supplementary Fig. 7c), consistent with previous results obtained in *Xenopus* oocytes[20,21]. However, similar as Deng et al. reported[21], T513 was not identified to be phosphorylated in our mass spectrometry analysis, either in the purified WT *At*SLAC1 alone protein or in the *At*SLAC1::OST1 fusion construct sample. Hitherto, the role of T513 remains elusive, and further studies are needed to reveal whether it is phosphorylated during *At*SLAC1 activation *in planta*, or the

enhanced currents for the T513E mutant are due to artificial local perturbations to the cytoplasmic region.

## Discussion

Activation of the guard cell anion channel SLAC1 has been identified to be essential to stomatal closure which helps plant to survive from environmental stresses, like high $CO_2/O_3$ concentrations, darkness, or drought. Phosphorylation of SLAC1 by multiple kinases has been extensively investigated, yet little is known about the structural mechanism how phosphorylation leads to channel opening. Previous studies mainly adopt the *Xenopus* oocyte, a simplified yet enlightening system, to examine the anion channel activity and reconstruct the activation process of SLAC1. However, it is not ideal for the recombinant protein expression to perform structural research. In this study, we used the HEK293 cell line, which is widely used for eukaryotic membrane protein expression, and successfully obtained the *At*SLAC1 protein in large quantity and good quality. Remarkably, when expressed alone in the HEK293 cells, *At*SLAC1 is phosphorylated by some endogenous kinase(s) and activated to a great extent. The electrophysiological results and phosphorylation sites identified by mass spectrometry suggest that it largely mimics the in vivo activation process as the previously identified key residues, S59, S86, and S120, are all found phosphorylated in the WT *At*SLAC1. The S59 site is required for *At*SLAC1 activation by the endogenous kinase(s) in HEK293 cells, as once mutated, the anion currents are much reduced. Notably, the anion current density for the phosphomimetic mutant, S59D, was less affected compared to the WT *At*SLAC1 when cells were pretreated with the kinase inhibitor, staurosporine, supporting a critical role of S59 site during channel activation (Supplementary Fig. 1g). This also suggests that the endogenous kinase(s) may have a preference of the phosphorylation sites similar to the $Ca^{2+}$-dependent CPK kinases in plants[20]. In addition, mass spectrometry analysis of the purified S59A mutant sample only identified the N-terminal S146 and the C-terminal S543 phosphorylated (Supplementary Fig. 2b), which is insufficient to activate the channel, consistent with previous reports[21].

Using the HEK293 cell line as a platform, we determined the *At*SLAC1 S59A mutant structure in an inactivated state. In the present structure, although the terminal tails are still partly invisible, the revealed regions clearly show that the N-terminus and C-terminus form a plug-like structure attached to the transmembrane domain, which blocks the anion pore and stabilized the channel in a closed state. Breaking the interactions between the plug and the transmembrane segments can noticeably activate the channel even in the absence of S59 site phosphorylation. Meanwhile, both the N- and C-terminus interact with the interhelical linker, $IH_{2-3}$, and destroying these interactions can also trigger channel opening. Residues located in the $IH_{2-3}$ have also been indicated to be required for the $CO_2$/bicarbonate sensing or acidosis-induced activation of the SLAC/SLAH anion channel family, like R256 in *At*SLAC1 and H330 in *At*SLAH3[25,26]. As shown by previous electrophysiological analysis, addition of $HCO_3^-$ further enhanced the anion currents of wild type *At*SLAC1 by OST1 phosphorylation, while the enhancement disappeared in the R256A mutant[26]. As our structure shows, R256 interacts with the I510 and I512 in the C-terminal tail and breaking this interaction triggers channel activation either in the presence or absence of S59 phosphorylation. The interaction of R256 with $HCO_3^-$ may neutralize its positively charged side chain and abolish the above interactions, thus leading to the channel activity enhancement. Protonation of H330 in *At*SLAH3 (corresponding to H260 in *At*SLAC1 in $IH_{2-3}$) and another histidine (H454 in *At*SLAH3) in the cytosolic

terminus of TM7 has been suggested to induce channel activation by switching SLAH3 from an inactivated dimer to an active monomer[25]. Although *At*SLAC1 cannot respond directly to acidosis, our structure of *At*SLAC1 and the high sequence similarities between *At*SLAC1 and *At*SLAH3, particularly in the cytosolic plug region, indicate that protonation of H330 in *At*SLAH3 may also affect the interactions between $IH_{2-3}$ and the cytosolic plug, and translate the conformational changes to channel activation.

For *At*SLAC1, *Bd*SLAC1 and the bacterial homolog *Hi*SLAC1, they all adopt a trimeric architecture, which may provide a stabilization of the protein in one way, or orchestration of multiple pores together in the other. Single-channel activity recordings of the S-type anion channels in guard cells (mainly contributed by SLAC1) have shown multiple conductance states and direct transitions among these states, suggesting cooperative openings of the oligomeric channels[27,28]. We checked the single-channel conductance of *At*SLAC1 in HEK293 cells. For the WT channel, which is partially activated, multiple states and direct transitions among these states were also observed, consistent with previous reports. We also checked the conductance for the R263A mutant, which releases the inhibitory effect of the cytosolic plug and has greatly enhanced anion currents as shown above, direct transitions among different states were more frequently observed, suggesting a strong cooperativity among subunits of the trimeric channel (Supplementary Fig. 7e, f). In another trimeric membrane protein in the *Arabidopsis*, the ammonium transporter (AMT1), an allosteric trans-activation mechanism has been proposed for its C-terminus, in which the C-terminus from one subunit interacts with the neighboring subunit and regulates its activity[29]. In our *At*SLAC1 structure, the cytosolic plug blocks the conducting pore from the same monomer. Based on the locations of the N- and C-termini forming the plug structure and the distance between two adjacent pores (Supplementary Fig. 7d), it is unlikely that the plug of one subunit also works on the neighboring subunit in the same trimer. Whether the cooperative channel opening is attributed to the rearrangements in the transmembrane domain upon activation or other unresolved elements in the cytosolic region, and the physiological relevance of the trimeric architecture of SLAC1 remain to be further investigated.

Taken the structural and electrophysiological information together, we propose a molecular mechanism for the SLAC1 activation by protein kinases (Fig. 5). Phosphorylation of SLAC1 in the N- and C-terminus leads to structural rearrangements in

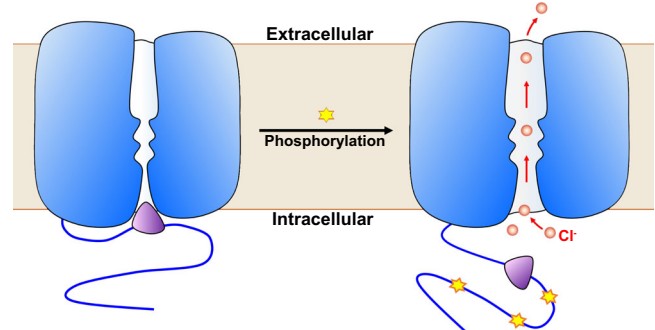

**Fig. 5 Cartoon model for SLAC1 activation by phosphorylation.** Phosphorylation of SLAC1 in the cytosolic domain formed by the N-terminal and C-terminal tails leads to structural rearrangements which triggers the plug region's dissociation from the transmembrane domain and releases its inhibition to the pore. Release of the cytosolic plug also induces structural rearrangement in the pore-forming helices, dilating the pore to allow anion permeation.

the cytosolic domain, triggers its dissociation from the transmembrane domain, especially for the plug-like structure, and releases its inhibition to the pore. The release of the cytosolic domain also induces structural rearrangement in the interhelical linkers and the pore-forming helices. Specially, the pore-constraining phenylalanine residues have to be rearranged to dilate the pore and allow anion permeation. The SLAC1 channels in *Arabidopsis*, rice and maize share high sequence identities (Supplementary Fig. 6). Notably, all the residues involved in the interactions between the cytosolic domain and the transmembrane domain and the key residues in $IH_{2-3}$ are identical in the crop plants, suggesting a similar regulation mechanism. In summary, the structural and functional characterizations of *At*SLAC1 reported here provide valuable insights into the activation process by phosphorylation and facilitate our understanding of stomatal regulation by the anion channel. They can also guide the future applications in agriculture by engineering and manipulations of the SLAC/SLAH family channels to obtain plants with enhanced resistance to the biotic or abiotic stress stimuli.

## Methods

**Protein expression and purification**. The DNA sequence of full-length *Arabidopsis thaliana* SLAC1 was synthesized with optimized codons by GENEWIZ and then cloned into the pCAG vector (Invitrogen) with the N-terminal Twin-Strep tag and Flag tag. All mutants were generated in the same way. The HEK293F cells (Sino Biological Inc.) at a density of $2 \times 10^6$ cells mL$^{-1}$ were transfected with plasmids of SLAC1. For 1 liter HEK293F cell culture, 1.5 mg SLAC1-S59A plasmids were premixed with 4 mg linear polyethylenimines (PEIs) (Polysciences) in 50 ml fresh medium for 15 min. The mixture was then added into cell culture followed by 15-min incubation. The transfected cells were cultured at 37 °C for 48–72 h before harvested by centrifugation at 2095 g for 10 min. Cell pellets were resuspended in the lysis buffer containing 25 mM HEPES, pH 7.4, 150 mM NaCl, 1.5% (w/v) DDM (Anatrace), and the protease inhibitor cocktail containing 1 mM phenylmethylsulfonyl fluoride (PMSF), aprotinin (1.3 mg ml$^{-1}$), pepstatin 3 (0.7 mg ml$^{-1}$) and leupeptin (5 mg ml$^{-1}$). After incubation at 4 °C for 2 h, the supernatant fraction isolated by centrifugation at 25,200 g for 1 h was collected and incubated with anti-FLAG M2 affinity gel (Sigma) at 4 °C for 30 min. The resin was rinsed three times, each with 10 ml of buffer containing 25 mM HEPES, pH 7.4, 150 mM NaCl, 0.06% (w/v) digitonin (Apollo Scientific). The protein was then eluted with wash buffer plus 200 μg ml$^{-1}$ FLAG peptides. The protein eluent was concentrated by a 100-kDa cut-off Centricon (Millipore) and further purified by Superose-6 Increase column (GE Healthcare) in the buffer containing 25 mM HEPES, pH 7.4, 150 mM NaCl, and 0.03% (w/v) digitonin. Finally, the peak fractions were concentrated to ~17 mg mL$^{-1}$ for single-particle cryo-EM sample preparation.

**Sample preparation and cryo-EM data acquisition**. For cryo-EM sample preparation, aliquots (4 μl) of the purified SLAC1-S59A was added to the glow discharged holey carbon grids (Quantifoil Au R1.2/1.3, 300 mesh), blotted with a Vitrobot Mark IV (ThemoFisher Scientific) using 4 s blotting time with 100% humidity at 8 °C, and plunged into liquid ethane cooled by liquid nitrogen. The grid was loaded into a Titan Krios (FEI) electron microscope operating at 300 kV, equipped with the BioQuantum energy filter with a 20 eV slit width and a K3 direct electron detector (Gatan). Images were recorded under a nominal magnification of 81,000 × using the EPU software in the super-resolution mode with a calibrated pixel size of 0.55 Å. Defocus values varied from −1.0 to −2.0 μm. Four thousand two hundred twenty-three image stacks were collected. Each stack was acquired with an exposure time of 3 s and dose-fractionated to 32 frames with a total dose of 50 e$^−$ Å$^{−2}$.

**Image processing**. A flowchart for the data processing is presented in Supplementary Fig. 3. Motion correction and dose weighting were performed using RELION 3.1's implementation of MotionCor2, and the stacks were binned twofold, resulting in a pixel size of 1.1 Å[30,31]. Subsequent data processing was all carried out with the twofold-binned micrographs. Defocus values were estimated by CTFFIND4[32]. After manually checked, 4,110 micrographs were selected particle picking. 3,079,765 particles were automatically picked using cryoSPARC (v.3.2.0)[33]. 1,017,886 good particles were selected after 2D classification. Further classifications were performed in cryoSPARC with a subsequent ab initio reconstruction into five classes with the imposed C1 symmetry. After non-uniform refinement in the C3 symmetry, an EM map at 3.3 Å was obtained with 569,306 particles. A set of 321,333 particles were selected with further heterogenous refinement and non-uniform refinement with the C3 symmetry, yielding a map with an overall resolution of 2.9 Å. Further CTF refinement and non-uniform refinement using cryoSPARC improved the resolution to 2.8 Å. A final round of Ab initio reconstruction into three classes in the C1 symmetry and the following non-uniform refinement in the C3 symmetry was performed. Finally, a map with an overall

resolution of 2.7 Å from 264,751 particles was achieved for the *At*SLAC1 S59A mutant. The overall resolution was estimated with the gold-standard FSC at a 0.143 criterion with a high-resolution noise substitution method[34,35]. Local resolution variations were estimated using ResMap[36]. A detailed data processing procedure can be found in Supplementary Fig. 3.

**Model building and refinement**. The 2.7 Å map for the *At*SLAC1 S59A mutant was used for de novo model building by COOT[37]. Bulky residues such as Phe, Tyr, Trp, and Arg were used to guide the sequence assignment, and the chemical properties of amino acids were considered to facilitate model building. Structure refinements were carried out by PHENIX in real space[38]. Overfitting of the model was monitored by refining the model in one of the two independent maps from the gold-standard approach and testing the refined model against the other map. Statistics of the 3D reconstruction and model refinement can be found in Supplementary Table 1. In the final structure model of the *At*SLAC1 S59A mutant, 368 amino acid residues were faithfully built and assigned, starting at residue 150 and ending at residue 517. The first 149 amino acids in the N-terminus and the last 39 residues in the C-terminus were unmodelled due to vague EM densities.

**Electrophysiology**. Whole cell patch-clamp recordings were performed on HEK293T cells co-transfected with *At*SLAC1 (WT or mutant) and pEGFP-N1. The cells were plated onto poly-L-lysine-coated coverslips 42 h after transfection and transferred to a patch-clamp recording chamber for the experiments 6 h later. Only GFP-positive cells were used for patch-clamping. Recordings were performed at room temperature using an Axopatch 200B (Molecular Devices) and a Digidata 1550B data acquisition System (Molecular Devices) under the control of pClamp software (Molecular Devices). Recording pipettes were made of borosilicate glass tubes using a P-1000 puller (Sutter Instrument). The filled pipettes have resistances of 8 ~ 10 MΩ. The bath solution for whole-cell voltage clamp recordings contained 150 mM HCl, 1 mM CaCl$_2$, 20 mM HEPES (pH adjusted to 7.4 with ~157 mM native form of NMDG). The pipette solution contained 150 mM HCl, 20 mM HEPES (pH adjusted to 7.4 with ~157 mM native form of NMDG). Liquid junction potentials were corrected for all recordings. When measuring the leakage of the seal, HCl in the bath solution was substituted with methanesulfonic acid. Recordings were carried out using a voltage step protocol (500 ms step pulses, from −100 mV to +100 mV, 20 mV step; $V_{holding} = 0$ mV) or a voltage ramp protocol (−100 mV to +100 mV in 1 s, every 10 s; $V_{holding} = 0$ mV). Single-channel currents were recorded using cell-attached mode. Statistics of the time constant (tau) values of channel open and close in single-channel recordings can be found in Supplementary Table 2. Data were analyzed using Clampfit and OriginPro.

**Reporting summary**. Further information on research design is available in the Nature Research Reporting Summary linked to this article.

## Data availability

The 3D cryo-EM density map of *At*SLAC1 S59A mutant has been deposited in the Electron Microscopy Data Bank under the accession number EMD-32633. Coordinates for the structure model have been deposited in the PDB under the accession code 7WNQ. Source data are provided with this paper. All other data are available from the corresponding author upon reasonable request.

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

## Acknowledgements

We thank the High-End Cryo-EM Platform, Core Facility Center for Life Sciences, University of Science and Technology of China, and the Electron Microscopy System at the National Facility for Protein Science in Shanghai (NFPS) for the EM facility support. We are grateful to all the staff members for their technical support on cryo-EM data collection. This work was supported by the Strategic Priority Research Program of Chinese Academy of Sciences (XDB37020103 to L.S.), the National Natural Science Foundation of China (31900885 to X.L., 31870732 to L.S.), the Natural Science Foundation of Anhui Province (2008085MC90 to X.L., 2008085J15 to L.S.), and the USTC Research Funds of the Double First-Class Initiative (YD9100002004 to L.S.). L.S. is supported by an Outstanding Young Scholar Award from the Qiu Shi Science and Technologies Foundation, and a Young Scholar Award from the Cyrus Tang Foundation.

## Author contributions

L.S. conceived the project. P.Z., C.X., X.Liu, C.C., and L.S. designed the experiments. Y.L. performed most of the molecular cloning, protein purification, and structure determination of *At*SLAC1. X.Li and Q.L. performed some of the molecular cloning and protein purification. Y.L. and Y.G. performed the cryo-EM data collection. X.Liu performed the cryo-EM data processing and model building. Y.D., L.Q., and C.C. performed the electrophysiology analysis of *At*SLAC1. All authors contributed to the data analysis and manuscript preparation. C.C., X.Liu, and L.S. wrote the paper.

## Competing interests

The authors declare no competing interests.
