## [Peer Review File · Nature Communications]

Structure of the Arabidopsis guard cell anion channel SLAC1 suggests activation mechanism by phosphorylationREVIEWER COMMENTS

Reviewer #1 (Remarks to the Author):

The authors present an inactive mutant of AtSLAC1 used generate an inactive cryoEM map. This was an enjoyable strait forward, manuscript, that provides additional insights to the cytoplasmic region of AtSLAC1, expanding on the knowledge base from previous SLAC1 structures. The manuscript was well written, and strait forward, with a clear story. The functional point mutations provide additional support to the authors conclusions that these newly resolved features work in concert to keep the channel in an inactive state. I did not have access to the pdb or mrcs during my review, so I cannot critically evaluate certain aspects relevant to these data.

I have the following technical comments that should be addressed to help improve accuracy, repeatability, and interpretation for the reader.

Although present in EDFig3, I would strongly urge the authors to also upload the half maps and FSC xml to onedep as these were absent in the validation report.

Although stated in the main text, I would advise revision of model building section to explicitly state what aa residue ranges could be modeled. Related, as EDFig5a is critical, to showing what your new structure brings to the table, if the authors can devise a way to make this more prominent it would be advantageous.

In the cryo-EM data acquisition section you have listed the use of a post-column GIF, but list no value for slit width used. Please revise to include, the slit width value used in data acquisition.

Additionally, was the camera used a K2 Summit or a K3, it is confusing as written?

The data were collected in super-resolution (SR) mode. Did the authors use binned or unbinned SR data for the final 3D refinement? Additional comments in the MotionCor2 section of image processing would be informative.

Also for data processing, was C3 symmetry imposed during all steps of 3D classification and refinement? It is unclear from the methods, please revise.

Comments:

Related, can the authors comment on the physiological relevance of the trimer?

Can the authors comment on the wt AtSLAC1 cryoEM data? As reported, it "could not be improved beyond 6A." I was curious if is there sufficient resolution about the general TMH arrangement from this map that provides any hints about the active state? Or perhaps comparison with other SLAC1 structures?

Can the authors comment on the pore lining phenylalanine conformational strain and comparison with BdSLAC1?

Minor

pg 18 line 374 inconsistent is one word

pg 13 lines 265,268 maybe helix breaker should be in quotes (i.e. "helix breaker") or a less colloquial term be used.

Pg 13 line 274 claims of plasma membrane trafficking are speculative

Reviewer #2 (Remarks to the Author):

This manuscript reports the structure of the closed state of the Arabidopsis SLAC1 channel. Interesting new findings are presented showing how the N-terminus and the C-terminus function in deactivating the channel by plugging the transmembrane domain. Furthermore, a number of interesting findings and conclusions are reported, that correlate with or provide mechanisms that could explain previous findings. Structural data further show that the inner part of the channel pore is constricted in the closed channel and provides possible mechanisms by which conformational changes induced by N terminal phosphorylation and also by C terminal removal can lead to channel opening. Overall, this interesting manuscript will be of broad interest to readers. Mostly minor revisions are recommended.

1. The model described on lines 371 to 372 suggesting first S59 needs to be phosphorylated before S86 and S120 are phosphorylated appears speculative and may be derived from data in HEK cells with an unknown native HEK cell kinase mediating activation. It is not clear if this applies to the plant cell environment. Clarifying this point, stating this more carefully, or adding data could be considered by the authors.

2. The methods for patch clamping refer to 150 mM HCl included in the bath solution and pipette solutions and addition of NMDG to neutralize the pH to 7.4 (lines 608-609). The final concentrations of NMDG should be included. Clarify whether these are in effect NMDG Cl salts.

3. As for other SLAC1 homologs, the Arabidopsis SLAC1 channel is shown to form three barreled trimers. Based on the structure, it should be discussed whether the data can exclude that the SLAC1 termini may regulate/plug neighboring channel pores in the trimer, as was shown for an ammonium transporter (Loqué D., Lalonde S. et al. Nature 2007). Since single channel analyses in guard cells suggested cooperative transitions between S-type single channel conductance levels (Schmidt C., Pl. Physiol. 1994), could the plug mechanism found here contribute to cooperativity among the subunits or is this less likely based on the structural data?

4. Lines 68-69 "either" and "or" does not reflect the model in which phosphorylation of SLAC1 is required "and" then this kinase-activated SLAC1 current can be enhanced. Reword sentence.

On lines 84 to 86 "either" and "or" does not reflect present findings. Delete "either" and replace "or" with "and by".

5. Page 8, line 181: replace "specifically" with "preferentially" and add "in oocytes".

6. Lines 218 to 220: Does the N terminus start (as stated) or end at residue 150? Similar question for the C terminus at residue 517.

7. Lines 60-63: Reference 4 reported on the rapid anion channels, not the S-type which was reported elsewhere.

8. The new structure is in an inactivated state (AtSLAC1 S59A). Perhaps an additional sentence or two on possible structural configuration mechanisms of channel activation could be added to the Discussion.

Reviewer #1:

We thank this reviewer for his/her constructive and helpful comments. S/he raised several specific points that are addressed below:

The authors present an inactive mutant of AtSLAC1 used generate an inactive cryoEM map. This was an enjoyable strait forward, manuscript, that provides additional insights to the cytoplasmic region of AtSLAC1, expanding on the knowledge base from previous SLAC1 structures. The manuscript was well written, and strait forward, with a clear story. The functional point mutations provide additional support to the authors conclusions that these newly resolved features work in concert to keep the channel in an inactive state. I did not have access to the pdb or mrcs during my review, so I cannot critically evaluate certain aspects relevant to these data.

I have the following technical comments that should be addressed to help improve accuracy, repeatability, and interpretation for the reader.

Major comments:

1. Although present in EDFig3, I would strongly urge the authors to also upload the half maps and FSC xml to onedep as these were absent in the validation report.

Point accepted. According to reviewer's suggestion, we uploaded all necessary files to onedep to get a detailed validation report which is submitted as a manuscript related file. In addition, we uploaded the PDB and MRC files for the structure model and EM map for the reviewer to get a full access to the detailed structure. We have also submitted the structure and map to PDB and EMDB database under the accession code of 7WNQ and 32633, respectively. And they have been officially approved by wwPDB with the status of "Hold for publication".

2. Although stated in the main text, I would advise revision of model building section to explicitly state what aa residue ranges could be modeled. Related, as EDFig5a is critical, to showing what your new structure brings to the table, if the authors can devise a way to make this more prominent it would be advantageous.

Point accepted. We have revised the model building section and explicitly stated the residue ranges built in the structure. The following description was added: "*In the final structure model of the AtSLAC1 S59A mutant, 368 amino acid residues were faithfully built and assigned,*

starting at residue 150 and ending at residue 517. The first 149 amino acids in the N-terminus and the last 39 residues in the C-terminus were unmodelled due to vague EM densities.”

Besides, we added a panel to Extended Data Fig.5 (panel 5a) to show clearly the extra elements firstly observed in our structure, and explained in the figure legend.

3. In the cryo-EM data acquisition section you have listed the use of a post-column GIF, but list no value for slit width used. Please revise to include, the slit width value used in data acquisition. Additionally, was the camera used a K2 Summit or a K3, it is confusing as written?

Point accepted. The slit width used was 20 eV. We have included this information in the revised manuscript (Page 33, Line 601). Sorry for the confusion, and the camera used was a K3 detector. We have corrected this error in the manuscript (Page 33, Line 601).

4. The data were collected in super-resolution (SR) mode. Did the authors use binned or unbinned SR data for the final 3D refinement? Additional comments in the MotionCor2 section of image processing would be informative.

Point accepted. The data was binned twofold during motion correction and dose weighting for subsequent data processing. Subsequent data processing was carried out with the twofold-binned micrographs. We include this critical information in the Image processing session in the revised manuscript (Page 33, Line 611-613).

5. Also for data processing, was C3 symmetry imposed during all steps of 3D classification and refinement? It is unclear from the methods, please revise.

Point accepted. During all steps of 3D classification, C1 symmetry was applied. For the refinement, C3 symmetry was applied. We have revised the manuscript according to this reviewer's suggestion.

6. Comments: Related, can the authors comment on the physiological relevance of the trimer?

Point accepted. This comment is also related to the third point raised by Review #2. In all the determined structures, SLAC1 forms a trimer. In one way, SLAC1 protein may be more stabilized in the trimeric architecture, and in another, the trimeric organization may affect the channel activity. As shown by the previous single-channel results (Schroeder J. I., et al.,

1993; Schmidt C., et al, 1994) and the single-channel results we have now obtained (added as Extended Data Fig.7e,f), there are cooperativities among monomers of the slow type anion channels. Although this cooperativity may not be contributed by the plug release alone (as discussed in the revised manuscript), subsequent conformation changes in the transmembrane region or other structural elements in one subunit upon channel activation may affect the opening of the pore in the neighbouring subunit. Thus, the trimeric organization of SLAC1 may synchronize multiple pores and boost the currents upon stimulations. In addition, another member of the SLAC channels, SLAH3, has been suggested to exist as a dimer in the inactivated state and turn to a monomer upon activation. Why the oligomeric states for SLAC1 and SLAH3 is different remains an open question, but this may suggest that the oligomeric state of SLAC channels may regulate the channel activities. Thus, in the revised manuscript, we have added a discussion about the trimer organization of SLAC1 and its possible physiological relevance (Page 20-21, Line 417-440).

7. Can the authors comment on the wt AtSLAC1 cryoEM data? As reported, it "could not be improved beyond 6Å." I was curious if is there sufficient resolution about the general TMH arrangement from this map that provides any hints about the active state? Or perhaps comparison with other SLAC1 structures?

We thank this review for this constructive suggestion. We did try with lots of efforts for the structure determination of the WT AtSLAC1. Unfortunately, for the WT AtSLAC1 map, the resolution did not allow us to faithfully assign all the transmembrane helices for further structural analysis or comparisons. In fact, we can only faithfully assign 3-4 TMHs in such a low-resolution map. Below shows a refined EM map for the WT AtSLAC1 with a reported resolution around 6.5 Å using cryoSPARC, which is the best map we could obtain. The S59A mutant structure is superimposed to this map. As we can see, densities corresponding to TM4-TM6 and TM8, most of which locate near the trimer interface, are relatively good for helix assignment. For the rest of the TMHs, the density was discontinuous or too vague to clearly assign. Although it may indicate that TMHs near the trimer interface may be more stable during channel opening, we cannot get further information about how the overall TMH rearranges. Thus, we did not discuss too much in our manuscript to avoid over interpretations.

8. *Can the authors comment on the pore lining phenylalanine conformational strain and comparison with BdSLAC1?*

Point accepted. Thanks for such a constructive comment. In the revised manuscript, we compared the structure details of the pore-lining phenylalanine residues and electrophysiological results for the alanine mutants. In the structures, the three phenylalanine residues discussed in our manuscript adopt almost identical configurations, suggesting conserved roles in channel gating. The electrophysiological results are also similar for *AtSLAC1* and *BdSLAC1*. While the conflicting results for the unphosphorylated and phosphorylated channels could not be well explained by the present data (especially due to lack of a structure of such channels in an activated conformation), these phenylalanine residues have to undergo structural rearrangements to open the channel.

According to this reviewer's suggestion, we add a panel in Extended Data Fig.5 (panel 5c) to show the structure comparison of the three pore-lining phenylalanine residues. A short discussion is also added in the revised manuscript (Page 13, Line 265-274).

Minor comments:

1. *pg 18 line 374 inconsistent is one word.*

Point accepted. Sorry for the typo. We have corrected two such typos in the revised manuscript.

2. *pg 13 lines 265,268 maybe helix breaker should be in quotes (i.e. "helix breaker") or a less colloquial term be used.*

Point accepted. "Helix breaker" is now in quotes. In addition, we added an explanation to helix breaker in the revised manuscript to make it easily understood.

3. *Pg 13 line 274 claims of plasma membrane trafficking are speculative.*

Point accepted. We have deleted such speculative description in the revised manuscript.

Reviewer #2:

We thank this reviewer for his/her constructive and helpful comments. S/he raised several minor points that are addressed below:

This manuscript reports the structure of the closed state of the Arabidopsis SLAC1 channel. Interesting new findings are presented showing how the N-terminus and the C-terminus function in deactivating the channel by plugging the transmembrane domain. Furthermore, a number of interesting findings and conclusions are reported, that correlate with or provide mechanisms that could explain previous findings. Structural data further show that the inner part of the channel pore is constricted in the closed channel and provides possible mechanisms by which conformational changes induced by N terminal phosphorylation and also by C terminal removal can lead to channel opening. Overall, this interesting manuscript will be of broad interest to readers. Mostly minor revisions are recommended.

1. The model described on lines 371 to 372 suggesting first S59 needs to be phosphorylated before S86 and S120 are phosphorylated appears speculative and may be derived from data in HEK cells with an unknown native HEK cell kinase mediating activation. It is not clear if this applies to the plant cell environment.

Clarifying this point, stating this more carefully, or adding data could be considered by the authors.

Point accepted. Thanks for this kind suggestion. Indeed, this is just the mass spec results got in HEK293 cells and situations in plant cell environment might be completely different. To avoid over- or mis- interpretation of the data, we carefully modified this description and deleted the speculative assumptions in the revised manuscript (Page 18, Line 379-382). Besides, to further validate that the activation of AtSLAC1 in HEK293 cells is through phosphorylation by endogenous kinases, we use a potent, broad spectrum kinase inhibitor to treat the cells before recordings. And the results show that the anion currents for the WT AtSLAC1 can be reduced by the inhibitor treatment. While for the S59D mutant, which mimics its phosphorylation state, the currents were merely affected (Extended Data Fig.1g).

2. The methods for patch clamping refer to 150 mM HCl included in the bath solution and pipette solutions and addition of NMDG to neutralize the pH to 7.4 (lines 608-

609). *The final concentrations of NMDG should be included. Clarify whether these are in effect NMDG Cl salts.*

Point accepted. We have included the critical information in the method section of the revised manuscript (Page 35, Line 656). The NMDG we use is in its native form, not Cl salt.

3. *As for other SLAC1 homologs, the Arabidopsis SLAC1 channel is shown to form three barreled trimers. Based on the structure, it should be discussed whether the data can exclude that the SLAC1 termini may regulate/plug neighboring channel pores in the trimer, as was shown for an ammonium transporter (Loqué D., Lalonde S. et al. Nature 2007). Since single channel analyses in guard cells suggested cooperative transitions between S-type single channel conductance levels (Schmidt C., Pl. Physiol. 1994), could the plug mechanism found here contribute to cooperativity among the subunits or is this less likely based on the structural data?*

Point accepted. Thanks for such a constructive suggestion. First, we checked the single channel conductance of AtSLAC1 in the HEK293 cell system. For the WT channel, which is partially activated, multiple states and direct transitions among these states were also observed. We also checked the conductance for the R263A mutant, which releases the inhibitory effect of the cytosolic plug and has greatly enhanced anion currents as shown above, direct transitions among different states were more frequently observed, suggesting a strong cooperativity among subunits of the trimeric channel, which further validate the previous reports (Extended Data Fig.7e-f). Second, in our SLAC1 structure, the plug works on the monomer of its own and based on the locations of the N-terminus and C-terminus of the plug structure and the distance between neighbouring channels, it is unlikely that the plug regulates neighbouring channel pores in the same trimer (Extended Data Fig.7d). Thus, the plug works differently with the ammonium transporter and the release of the plug alone may contribute little to the cooperativity. The rearrangements in the transmembrane domain or other unresolved structural elements in the cytosolic domain may contribute more to the cooperativity. Thus, we added a discussion in the revised manuscript of such scenarios and comparison with the ammonium transporter (Page 20-21, Line 419-440) and three panel in the Extended Data Fig.7 (panel 7d-f).

4. *Lines 68-69 “either” and “or” does not reflect the model in which phosphorylation of SLAC1 is required “and” then this kinase-activated SLAC1 current can be enhanced.*

*Reword sentence. On lines 84 to 86 “either” and “or” does not reflect present findings.
Delete “either” and replace “or” with “and by”.*

Point accepted. Both sentences have been revised according to this suggestion.

5. Page 8, line 181: replace “specifically” with “preferentially” and add “in oocytes”.

Point accepted. Revised accordingly.

*6. Lines 218 to 220: Does the N terminus start (as stated) or end at residue 150?
Similar question for the C terminus at residue 517.*

Point accepted. We are sorry for the confusion. In the structure model, the N terminus starts at residue 150 and the C terminus ends at residue 517. The first 149 and the last 39 residues of the protein were unmodelled due to vague EM densities. We revised the sentence accordingly in the manuscript (Page 11, Line 222-224).

*7. Lines 60-63: Reference 4 reported on the rapid anion channels, not the S-type
which was reported elsewhere.*

Point accepted. Sorry for the misuse of this reference. We have deleted this reference in the revised manuscript.

*8. The new structure is in an inactivated state (AtSLAC1 S59A). Perhaps an
additional sentence or two on possible structural configuration mechanisms of
channel activation could be added to the Discussion.*

Point accepted. In the discussion, we have proposed a model for the SLAC1 activation mechanism upon phosphorylation. We make it more clearly now in the revised manuscript according to the reviewer's comment (Page 21, Line 444-450). Description of the model and possible structural configuration is as following: “Phosphorylation of SLAC1 in the N- and C-terminus leads to structural rearrangements in the cytosolic domain, triggers its dissociation from the transmembrane domain, especially for the plug-like structure, and releases its inhibition to the pore. The release of the cytosolic domain also induces structural rearrangement in the interhelical linkers and the pore-forming helices. Specially, the pore-constraining phenylalanine residues have to be rearranged to dilate the pore and allow anion permeation.”

REVIEWERS' COMMENTS

Reviewer #1 (Remarks to the Author):

The authors have addressed my previous comments. I have looked at the map and models and they seem adequate and agree with figures.

A minor comment:

pg 12 last sentence Together... confusing - revise in proofs

Reviewer #2 (Remarks to the Author):

The authors have addressed my previous comments. This manuscript reports a number of interesting findings linked to the resolution of the structure of the closed state of the Arabidopsis SLAC1 channel. Interesting new findings are presented showing how the N-terminus and the C-terminus function in deactivating the channel by plugging the transmembrane domain. One minor previous comment remains to be addressed.

Reword Lines 67 to 71: "either" "or" does not reflect present models: Both models have been proposed to work together presently for (1) phosphorylation-dependent channel activation and (2) enhancement of activity. "and" would better reflect the present state of knowledge.

This is an important advance with many implications for plant anion channel regulation.

Reviewer #1:

The authors have addressed my previous comments. I have looked at the map and models and they seem adequate and agree with figures.

A minor comment:

pg 12 last sentence Together... confusing - revise in proofs

Point accepted. We have revised the sentence as below to avoid confusion:

“Combining F450A with the phosphorylation site mutation S59A, the S59A/F450A double mutant increased the inward current density by almost two folds compared to the S59A single mutant, while still remained much lower than the WT AtSLAC1 (Fig.3c).”

Reviewer #2:

*The authors have addressed my previous comments. This manuscript reports a number of interesting findings linked to the resolution of the structure of the closed state of the Arabidopsis SLAC1 channel. Interesting new findings are presented showing how the N-terminus and the C-terminus function in deactivating the channel by plugging the transmembrane domain. **One minor previous comment** remains to be addressed.*

Reword Lines 67 to 71: "either" "or" does not reflect present models: Both models have been proposed to work together presently for (1) phosphorylation-dependent channel activation and (2) enhancement of activity. "and" would better reflect the present state of knowledge.

This is an important advance with many implications for plant anion channel regulation.

Point accepted. We have revised the description in the manuscript as this reviewer suggested:

"HCO₃⁻ can lead to the activation of multiple protein kinases that function in parallel, like the SnRK2 (Snf1-related protein kinase 2) family member OST1 or the LRR-RLK (leucine-rich repeat receptor like protein kinase) family member GHR1, which phosphorylates SLAC1 and somehow opens the channel. Meanwhile, it also binds directly to the transmembrane domain of SLAC1 and enhance the channel activity."

This would better reflect both models as we now understand.